# Pandemic COVID-19 Influence on Adult’s Oral Hygiene, Dietary Habits and Caries Disease—Literature Review

**DOI:** 10.3390/ijerph191912744

**Published:** 2022-10-05

**Authors:** Aleksandra Wdowiak-Szymanik, Agata Wdowiak, Piotr Szymanik, Katarzyna Grocholewicz

**Affiliations:** 1Department of Interdisciplinary Dentistry, Pomeranian Medical University, 70-111 Szczecin, Poland; 2Department of Nuclear Medicine, Charite Universitatsmedizin, 13353 Berlin, Germany; 3DentalPort Szymanik Dental Office, 72-600 Świnoujście, Poland

**Keywords:** pandemic, COVID-19, SARS-CoV-2, oral hygiene, dental care, dietary habits, caries diseases, eating habits

## Abstract

Coronavirus disease (COVID-19) is an infectious disease caused by the SARS-CoV-2. The pandemic over the past two years has completely changed people’s daily habits with an impact on oral hygiene, eating habits, and oral health. Materials and methods: The available literature was reviewed on the PubMed platform and from other sources MEDLINE and Cochrane Reviews. The analysis included comparative and clinical trials as well as pragmatic clinical/randomized controlled trials, and observational studies which focused on the effects of COVID-19 on the dietary habits of the population, oral hygiene, and caries incidence. Results: The analysis shows that the COVID-19 pandemic had a significant negative impact on dietary habits and an ambiguous impact on oral health habits of the population. The researchers showed that patients’ visits were limited only to those necessary, because of fear of infecting with the SARS-CoV-2. Conclusions: The literature review shows that the COVID-19 pandemic, by affecting many aspects of everyday life, including eating habits, caring for oral hygiene, and avoiding regular visits to the dentist, may generate an increase in oral diseases. Due to the differences in knowledge results, further research observations in this field are necessary.

## 1. Introduction

Coronavirus disease (COVID-19) is an infectious disease caused by the SARS-CoV-2 virus, that affects mainly the respiratory tract and can lead to serious complications such as kidney failure. The virus can spread from an infected person’s mouth or nose in small liquid particles when they cough, sneeze, speak or breathe (WHO 2020) [1]. It was declared a pandemic by the World Health Organization in March 2020. Since then, mask-wearing, hand hygiene, and physical distancing are essential to preventing COVID-19.

Oral health is an important component of health and overall well-being. It is estimated that oral diseases affect nearly 3.5 billion people and according to the Global Burden of Disease 2019, untreated dental caries in permanent teeth, with a total of two billion people suffering, is the most common health condition (GBD 2019) [2]. Dental caries is localized destruction of dental hard tissues instigated by acidic by-products from bacterial fermentation of carbohydrates (Selwitz et al. 2007) [3]. Since dental caries does not progress without the bacteria present in dental plaques, daily plaque removal by brushing, flossing, and rinsing is one of the best ways to prevent dental caries and periodontal disease. The enhancement of dental plaque accumulation induces a rising of caries and periodontal diseases, which lead to inflammation of soft and hard tissues, intensifying alveolar bone loss, and finally earlier teeth loss [4,5]. Proper brushing and flossing methods may be taught at the dental office during routine check-ups [6]. During the past two years, the global population had to completely change their daily habits and during multiple lockdowns, people started working from home and stopped going out as often. As a result, physical activity and social contacts were also drastically limited. COVID-19 Pandemic was strongly connected with social isolation. Stress, anxiety, and loneliness as well as wearing masks daily were important factors, that impacted the health of the global population. These are associated with multiple health-risk behaviors such as having an unhealthy diet, high alcohol intake, and higher smoking frequency [7]. Changes in people’s dietary habits such as increased sweets consumption and worse oral hygiene, could possibly result in an increase in the prevalence of caries diseases. It is impossible to find the actual statistic that compares the year 2019 with the year 2022.

This literature review aims to show, how COVID-19-Pandemic and the resulting lockdown restrictions influenced adults’ oral hygiene, dietary habits, and caries diseases.

## 2. Materials and Methods

### 2.1. Search Strategy

Carrying out a systematic electronic search in MEDLINE and Cochrane Reviews using PubMed enabled relevant publications to be identified. The database was searched in the time from 1 April 2022 to 3 April 2022. All publications from 2020 to 2022 were considered. Keywords and any variations thereof could be identified using the standardized ‘MeSH’ (Medical Subject Heading) system. The following terms were used:

(COVID-19 OR SARS-CoV-2 OR Pandemic) AND (Oral Hygiene OR dental care OR dental OR Dietary habits OR Caries OR Eating Habits OR Dentistry)

As a part of the extended research, the references of Lages et al. were scanned and through snowballing effect, no more studies, that meet our inclusion criteria, could be identified.

The articles found were read and either considered or rejected for the review, according to the inclusion and exclusion criteria. (see Figure 1).

### 2.2. Inclusion and Exclusion Criteria

Inclusion criteria were observational studies on the human population (adults only), published in the years 2020–2022 in English, that addressed the issue of COVID-19 Influence on the population´s dietary habits, oral hygiene, and changes in the prevalence of caries, were included in the systematic review.

In studies in which COVID-19 was not mentioned as an impacting factor and in which children were involved, dentistry and other review subjects were not addressed, and were excluded, as were one-case reports and other systematic reviews.

### 2.3. Data Extraction

To assess the studies systematically, they were reviewed regarding the inclusion and exclusion criteria as well as characteristics of the study population, changes in diet and oral hygiene, which resulted in changes in the prevalence of caries diseases and how the examined parameters were measured and the main findings for each study as well as its limitations.

### 2.4. Assessment of Quality of Study

Many factors have an influence on the quality of the studies, including possible confounders and limitations. All studies included in the systematic review had cross-sectional characteristics. A relative bias has the potential to skew results and for estimation of its risk, the Joanna Briggs Institute (JBI) Critical Appraisal Checklist for analytical cross-sectional study was used. The maximum possible score for each study was 8. Because of the self-reported data character in every study, none of them obtained the maximum score—the main variables were not measured in a valid and reliable way, only through subjective observation of each responder, which correlates with a high risk of confounding and bias. Other criteria can be seen in Table 1. Most studies have mentioned its limitations but only two of them have directly identified possible confounders- these studies: Paltrinieri and Maestre obtained the highest score of 7. The lowest score of 4 had three studies: Paszyńska, Faria, and Ferrante, due to a lack of attention to possible confounders and limitations [9,10,11,12,13]. (Table 2).

## 3. Results

### 3.1. Search Results

The systematic MEDLINE (PubMed) search resulted in 1370 publications being identified, the Cochrane Library search resulted in 113 publications being identified. After reviewing their abstracts, 48 articles met the inclusion criteria. Upon reviewing 48 full texts, eleven studies were excluded for the following reasons: seven results of the studies were irrelevant for this systematic review, because the important review subjects were not mentioned, in two studies children were involved and the focus of two studies was weight management which was also one of the exclusion criteria of this systematic review, which led to a total of eleven included studies. For an overview of the selection process see Figure 1.

### 3.2. Characteristics of the Studies

The most important methodological characteristics of the included studies as well as the features of the study population are summarized in Table 3. All publications had a cross-sectional design, Souza [15] described their study as an observational study and Cicero [16] described their study as a Sub-study of a longitudinal population study. All data has been collected in 2020, except Paszynska [11], which collected their data in 2021. Except for Paszynska and Cicero where the Surveys were collected at the vaccination point and through a phone interview, the questionnaires were distributed online and answered by internet users, and all answers were self-reported [11,16].

### 3.3. Attributes of the Studies

Important attributes of the involved studies are shown in Table 4. All studies employed the use of questionnaires or scales to measure subjective feelings, amount of eaten food, and frequency of activities such as tooth brushing or smoking. More studies [9,10,11,14,15,16,18,19] were investigating changes in eating habits than changes in oral hygiene habits [17,18,19].

### 3.4. Impact of COVID-19 on Dietary Habits

Six [9,10,11,12,14,16] of the included studies showed changes in people’s dietary habits during the COVID-19 Pandemic, but only one of them [10] did not mentioned changes in alcohol consumption and smoking frequency. Every single study reported a significant increase in eating frequency with sweet snack preference, which in this case means worsening of eating habits. In the study of Paltrinieri, 1/3 of the population reported paying more attention to eating healthy, which cannot be objectively assessed [9].

One study [15] focused only on changes in alcohol consumption and smoking frequency.

Almost every single study showed that during Quarantine alcohol consumption as well as smoking frequency increased significantly. Paltrinieri [9] came to the conclusion, that changes in alcohol drinking occurred in both directions equally. Research by Souza [15] pointed out that an increase in frequency does not always mean an increase in dosage. In this research, responders reported a higher frequency of alcohol consumption but a reduction in the dose. In the study of Cicero [16], quarantine did not significantly modify smoking habits. Caramida [19] reported an increased number of non-smokers (7%). In the study of Paszynska [11], almost 82% of responders consumed alcohol during quarantine.

### 3.5. Impact of COVID-19 on Oral Hygiene Habits

Not a single study that investigated the effect of COVID-19 on the change in the prevalence of caries could be identified. Only four studies [17,18,19] mentioned changes in oral hygiene, and two of them mentioned also dietary habits [18,19]. Increased fear of visiting the oral health professional among the respondents was shown and the reason for that was the pandemic and the fear of possible infection with the virus. Due to tooth pain, 10% of responders had to seek emergency dental care (Pinzan-Vercelino [17]). In the research of Pinzan-Vercelino and Faria, an increase in bad breath awareness could be observed [12,17]. A total of 15% of individuals started considering having bad breath due to mask wearing, which resulted in a higher frequency of tooth brushing among this group. Due to mask-wearing, a significant number of subjects (6.8%) were also less concerned about their smile and oral hygiene, which resulted in the opposite effect. A study by Faria reported that 24% of individuals increased the frequency of toothbrushing [12]. On the contrary, Caramida reported no significant change in the frequency of toothbrushing, but the time spent on toothbrushing increased significantly (5.2%) among the medical professionals’ group [19]. Responders in a Sari study brushed their teeth 41.1% more regularly and 32.49% had their brushing routine disrupted [18].

## 4. Discussion

COVID-19 has a great impact on people’s life in general. To minimize the spread of coronavirus, the global population had to adapt to living in lockdown for several months. During this time daily habits were forced to change drastically. This systematic review investigated the impact of COVID-19 on adults’ dietary and oral hygiene habits. Studies assessing the influence of the global pandemic on the prevalence of caries diseases or caries parameters could not be identified. Caries is linked to unhealthy dietary and oral hygiene habits, and as the most common dental disease, there is a need to investigate COVID-19’s influence on its prevalence more.

There is a significant correlation between COVID-19 and the worsening of peoples’ eating habits. In most included studies, the worsening of populations’ eating habits could be observed. Responders were consuming much more unhealthy products with the preference for sweets, also a higher frequency of alcohol and cigarette consumption was reported. These products are very likely to be some sort of trigger for caries disease, for patients that do not take care of their oral hygiene. But not only worse eating habits could possibly influence the spreading of caries. What could be shown in this systematic review is, that because of daily mask-wearing, patients with halitosis were more aware of their bad breath, which resulted in higher tooth brushing frequency. On the other side, people were less concerned about their teeth esthetics, and they reduced their brushing frequency. Thus, masks are one factor that caused a reduction of caries risk in one group as well as the increase of caries risk in the other group of responders. As the study of Caramida shows the background of the responders (their education level, and cultural habits) strongly correlates with oral hygiene awareness [19]. There is a lacking awareness in some social groups. Through the education of this population, positive changes in their behavior could be made. The review of Al-Bayaty F. H. et al. [20] points out the importance of adequate oral care for critically ill patients with COVID-19. It is hoped, that through reduction of dental plaque, for example using 0.12%-chlorhexidine, the risk of atypical pneumonia and/or other infections involving distant organs will be greatly reduced. This conclusion supports the possibility of using oral hygiene among healthy patients as a possible tool to prevent the spreading of the infection.

Due to the lockdown, regular dental check-ups have also suffered. Fear of possible infection at the oral health professional office resulted in patients delaying treatment. Sari reported 50.4% of patients hesitated to go to the dentist and a total of 75.6% of responders thought that dental clinics were at risk of COVID contamination [18]. It means that because of possible contamination risk, a significant number of responders delayed their dental appointment which may result in the progression of caries disease among the patients. Only one study that also focused on the association between social isolation and oral health status could be identified.

The research of Paradowska shows that there are many oral symptoms of COVID-19, but the coexistence with the main disease has not been fully stated and understood [21]. The study of Nuno-Gonzalez A. et al. suggests that oral cavity findings are present in 25.65% of cases [22]. The common symptoms are blisters, ulcerations, and desquamative gingivitis as well as ulcerations of the tongue. Those symptoms cause patients’ pain and discomfort, which potentially could be the reason to reach out to the specialists more often, which in the case of COVID, was not possible, because patients with acute COVID infection were obligated to stay home. In addition to that, the review of Lewandowska M. et al. shows, that due to increased fear of pandemic among dentists, many practices were closed as well as the quality of dental service decreased, making it difficult for patients to access the treatment [23].

In the study by Berberoglu B. et al., the increase in dental anxiety among females could be triggered by the pandemic [24]. However, it could affect both genders. Toothache was patients’ main reason to seek emergency care. In our systematic review, the anxiety of potential infection was also the reason for a delayed dentist appointment. The patients reached out first because of severe pain. In contrast to Lages et al., this systematic review could identify an association between pandemic-resulting lockdown and oral health status [25]. As well as in the systematic review by Lages et al., this research also could not find any data directly describing changes in caries prevalence and caries parameters during the pandemic [25]. Both reviews show the need to conduct more research in this field. Also, they both are emphasizing the importance of oral hygiene and a healthy diet to maintain good oral health.

## 5. Conclusions

The MeSH search system proved to be very useful in identifying publications relevant to this review. However, studies that are not yet published cannot be found in electronic databases. Database on responders’ self-reported habits that possibly could be over/under-estimated, partly forgotten, and therefore not completely according to the truth.

## 6. Limitations

This systematic review has potential limitations. All studies included in this systematic review had a cross-sectional design with a high risk of bias. Most data were collected in 2020, when the pandemic just started, so the living conditions could vary in comparison to the latest phase of the pandemic in 2022.

## Figures and Tables

**Figure 1 ijerph-19-12744-f001:**
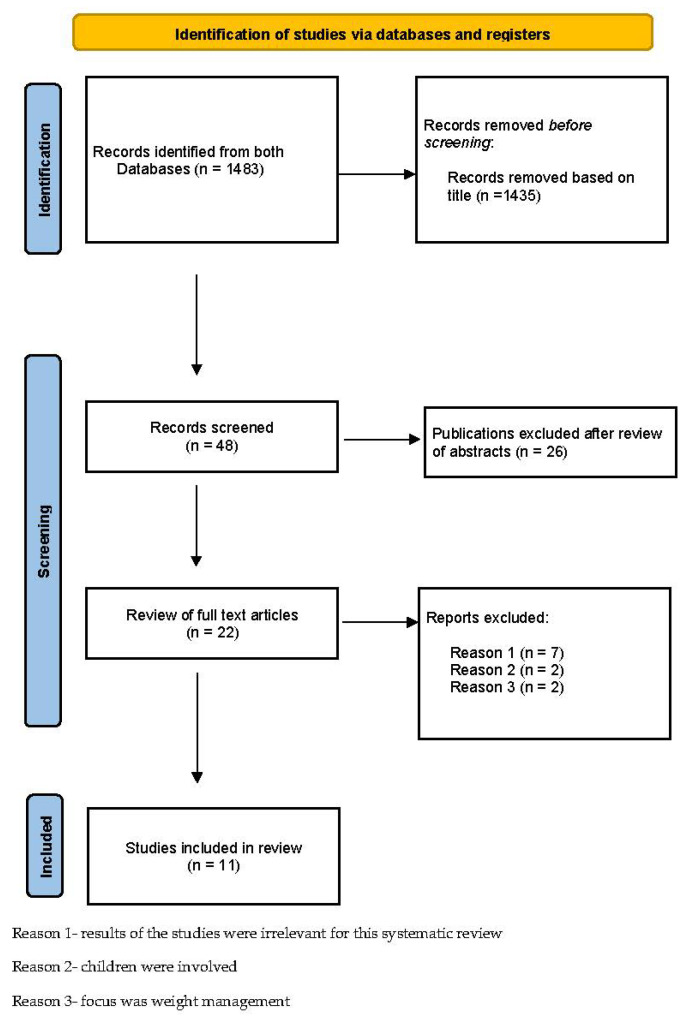
Identification of chosen studies via database registers [8].

**Table 1 ijerph-19-12744-t001:** JBI Critical Appraisal Checklist for Analytical Cross-Sectional Studies.

The Joanna Briggs Institute (JBI) Critical Appraisal Checklist for Analytical Cross-Sectional Study				
Major Components	Response options			
1. Were the criteria for inclusion in the sample clearly defined?	Yes	No	Unclear	Not applicable
2. Were the study subjects and the setting described in detail?	Yes	No	Unclear	Not applicable
3. Was the exposure measured in a valid and reliable way?	Yes	No	Unclear	Not applicable
4. Were objective, standard criteria used for measurement of the condition?	Yes	No	Unclear	Not applicable
5. Were confounding factors identified?	Yes	No	Unclear	Not applicable
6. Were strategies to deal with confounding factors stated?	Yes	No	Unclear	Not applicable
7. Were the outcomes measured in a valid and reliable way?	Yes	No	Unclear	Not applicable
8. Was appropriate statistical analysis used?	Yes	No	Unclear	Not applicable
Total	8 points			

**Table 2 ijerph-19-12744-t002:** Analysis of articles based on JBI Critical Appraisal Checklist for Analytical Cross-Sectional Studies.

Study	Ferrante G. et al., 2020 [12]	Skotnicka M. et al., 2021 [13]	Maestre A. et al., 2021 [9]	Souza TC. et al., 2021 [14]	Paltrinieri S. et al., 2021 [8]	Cicero, A. et al., 2021 [15]	Pinzan-Vercelino, C. et al., 2020 [16]	Sari a. et al., 2021 [17]	Faria SFS et al., 2021 [11]	Paszynska, E. et al., 2021 [10]	Cărămidă M. et al., 2022 [18]
1	No	Yes	Yes	Yes	Yes	Yes	Yes	Yes	No	Yes	Yes
2	Yes	Yes	Yes	Yes	Yes	Yes	Yes	Yes	Yes	Yes	Yes
3	No	No	No	No	No	No	No	No	No	No	No
4	Yes	Yes	Yes	Yes	Yes	Yes	Yes	Yes	No	Yes	Yes
5	No	No	Yes	No	Yes	No	No	No	Yes	No	No
6	No	No	No	No	Yes	No	No	No	No	No	No
7	Yes	Yes	Yes	Yes	Yes	Yes	Yes	Yes	Yes	Yes	Yes
8	Yes	Yes	Yes	Yes	Yes	Yes	Yes	Yes	Yes	Yes	Yes
Total	4/8	5/8	6/8	5/8	7/8	5/8	5/8	5/8	4/8	5/8	5/8

**Table 3 ijerph-19-12744-t003:** The most important methodological characteristics of the included studies.

Author, Year, Country	Design	Duration of Study	Population	Age of Study Group, Sex	Number of Participants
Ferrante G. et al., 2020, Italy [13]	Cross-sectional study	21 April 2020–7 June 2020	Internet responders	14–70+, female and male	7847
Skotnicka M. et al., 2021, Poland [14]	Cross-sectional study	1 October 2020–30 October 2020	Internet responders	>18, Female and male	1071
Maestre A. et al., 2021, Spain [10]	Cross-sectional study	1 April 2020–4 May 2020	Internet responders	>18, Female and male	1640
Souza TC. et al., 2021, Brazil [15]	Cross-sectional study	from August 2020 to September 2020	Internet responders	>18, Female and male, excluding pregnant women	1368
Paltrinieri S. et al., 2021, Italy [9]	Cross-sectional study	4 May 2020–15 June 2020	Internet responders	>18, Female and male	1826
Cicero, A. et al., 2021 Italy [16]	Sub-study of a longitudinal population study	February 2020–April 2020	Phone Interview Responders	>18, Female and male	359
Pinzan-Vercelino, C. et al., 2020, Brazil [17]	Cross-sectional study	10 June 2020–20 June 2020	Electronic survey (Google Forms)	>18, Female and male, wearing face masks in the last 30 days	1346
Sari a. et al., 2021, Turkey [18]	Cross-sectional study	1 August 2020–1 October 2020	Online survey via email/WhatsApp	>18 and <65, Female and male	1227
Faria SFS et al., 2021, Brazil [12]	Cross-sectional study	August 2020	Email questionnaire	Members and staff of Federal University of Minas Geiras	4647
Paszynska, E. et al., 2021, Poland [11]	Cross-sectional study	March 2021–May 2021	Self-designed questionnaire conducted at the COVID-19 Vaccination point	>18, Female and male, COVID vaccinated individuals	2574
Cărămidă M. et al., 2022, Romania [19]	Cross-sectional study	May 2020	Questionnaire distributed via digital platforms	18–75, Medical professionals and the general population	800

**Table 4 ijerph-19-12744-t004:** The attributes of the involved studies.

Author	Aim of the study	Form	Tools	Variables	Results	Limitations
Ferrante G. et al. [13]	to investigate the impact of COVID-19 on daily habits	-25 questions about eating and smoking habits	online self-report questionnaire	-nutrition -alcohol intake - smoking	-among smokers 30% reported an increase in the number of cigarettes smoked per day -the increase in alcohol consumption was reported by 17.3% of respondents -the number of responders who increased sweets intake (45%) is almost three-fold that of those who reduced them (16.5%)	-lack of representativeness of the considered sample -all data collected are self-reported = not completely reliable -small number of interviews (14%) was collected after May 3^rd^, the date marking the end of the most rigid period of social isolation
Skotnicka M. et al. [14]	to examine changes in dietary habits during COVID-19	- 0 questions about dietary habits	online self-report questionnaire	-sweets and snacks, juice and sweets drinks consumption -alcohol intake	-5.61% increase of sweets consumption -9.81 % increase of alcohol consumption -0.65% decrease of juice and sweet drinks consumption	-were not mentioned
Maestre A. et al. [10]	to identify the main changes in the eating habits during COVID-19	Modified Food consumption frequency questionnaire (FFQ) consisting of 18 questions	online self-report questionnaire	-sweets and candy consumption -snacks consumption -sugary sodas intake	Consumption’s Increase of: -sweets and candy = 32.7% -Snacks = 5.9% -sugary sodas = 28.2% --- > worsening of the dietary patterns of the population with an increase in the frequency of consumption of snacks and products rich in sugars	-telematic sampling system (over- and under-estimation possibility) -all data collected are self-reported = not completely reliable -more than 65% of the sample belonged to the Valencian Community- lack of representativeness of the considered sample -BMI biases possible
Souza TC. et al. [15]	to assess changes in food choices of adult Brazilians before and during the COVID-19 pandemic	Modified Food consumption frequency questionnaire (FFQ) consisting of 18 Questions	online self-report questionnaire	-alcohol consumption -frequency of smoking	-increase in the frequency of consumption of alcoholic beverages, but a reduction in the dose -increase in the frequency of smoking, but no significant difference in the number of cigarettes smoked per day	-all data collected are self-reported = not completely reliable -habits before the pandemic could possibly not be exactly remembered
Paltrinieri S. et al. [9]	to describe the changes in diet, alcohol drinking, and cigarette smoking during lockdown	-49 questions about dietary habits	online self-report questionnaire	-sweets and candy consumption -snacks consumption -sugary sodas intake -alcohol consumption -frequency of smoking	-diet changes in 17.6% of cases were for the worse (eating more snacks, sweets, carbonated drinks), in 33.5% improved (paying more attention to eating healthier) -in alcohol drinking changes occurred in both directions equally, since 12.5% of individuals increased their alcohol consumption and 12.6% decreased it -7.7% of smokers reported an increase and 4.1% a decrease in cigarette smoking	-sample was not representative of the resident population -self-perceived phenomena -age, sex, and education level as possible sources of bias
Cicero A. et al. [16]	to evaluate the effect of COVID-related quarantine on smoking and dietary habits	The Dietary Quality Index (DQI), a validated tool providing information on the usual food intake of 18 food items, grouped into three food categories	phone Interview	-sweets and sugar intake -alcoholic drinks intake -smoking frequency	-during quarantine, the interviewed subjects significantly increased the consumption of simple sugars, added fats, and alcohol, while overall increasing the carbohydrates and fat intake -quarantine did not significantly modify smoking habit (2.2% reduced their habit, 1.7% increased smoking) of respondents	-sample was not representative of the resident population -self-perceived phenomena -age, sex, and education level as possible sources of bias
Pinzan-Vercelino, C. et al. [17]	to evaluate the impact of the use of face masks during the COVID-19 pandemic on oral hygiene habits	41 multiple-choice questions oriented in the oral hygiene and oral conditions self-perception directions	online self-report questionnaire	-halitosis -need to seek dental care -concern about smile esthetic -frequency of teeth brushing	-10% needed to seek emergency dental care mainly due to tooth pain - the number of subjects with no concern about smile esthetics increased significantly (6.8%) -women, younger people, and subjects that had completed high school or had university or professional degrees reported more concern with smile esthetics - the subjects were brushing their teeth fewer times per day and people are less concerned about oral hygiene -the number of subjects that reported having halitosis increased significantly and there was a significant association between toothbrushing less time per day	-observations could not be directly inferred for other populations
Sari a. et al. [18]	to investigate the effects of COVID-19 on oral health status	total of 24 mandatory closed-ended questions	online self-report questionnaire	-frequency of teeth brushing -usage of oral care products -consumption of sugary foods -visiting dental clinics	-41.1% started brushing more regularly and 32.49% of brushing routines were disrupted -9.9% started using oral care products more regularly, while 7.3% more irregularly -33.3% increase in sugary food consumption -50.4% hesitated to go to the dentist compared to before COVID-19, and a total of 75,6% thought that dental clinics were at risk of COVID contamination.	-survey required the use of smartphones, therefore mostly young people mostly with high economic status participated -cross-sectional design
Faria SFS et al. [12]	to assess self-reported halitosis and oral hygiene habits with the wearing of face masks during the pandemic	questionnaire included a total of 18 items	email self-report questionnaire	-considering having a bad breath -changes in oral hygiene habits -seeking a healthcare professional	-14% of individuals started to consider having bad breath -24% increased frequency of toothbrushing, 5.8% of mouthwa use. Mainly individuals, who realized their bad breath significantly changed their oral hygiene -only 0,4% were seeking forsh a healthcare professional	-questionnaire-survey study with bias possibilities -most responders were female and limited to the university staff and students
Paszynska, E. et al. [11]	to assess whether the COVID-19 pandemic affected dietary choices, oral hygiene habits, and willingness to visit the dental office	questionnaire- number of questions not mentioned	self-report questionnaire	-eating frequency -smoking and alcohol consumption	-13.4% declared increased eating frequency with 19.1% sweet snack preference -only 17.3% did not report any alcohol consumption -3.4% declared smoking more than one pack a day -only 52.9% had a dental visit in 2020 -25% were afraid to schedule a dental examination	-limited access to some societal groups -limited frame of the questionnaire
Cărămidă M. et al. [19]	to assess differences in oral hygiene routine, smoking, and eating habits during the lockdown	online questionnaire formed of 17 items	self-report questionnaire	-frequency of toothbrushing -use of dental floss -smoking frequency -eating habits	-frequency of toothbrushing did not change, however, the time spent on toothbrushing increased in the medical professionals’ group significantly (5.2%), as well as the frequency of dental floss usage (3.5%) -sweet snacks consumption increased significantly in both groups (ca. 7%) -number of non-smokers increased in both groups (ca. 6%)	-self-reported habits with the possibility of under- or overestimation - not fully representative of the general population -only first quarantine context

## Data Availability

Not available.

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
