# Peer review of "Pandemic COVID-19 Influence on Adult’s Oral Hygiene, Dietary Habits and Caries Disease—Literature Review"

_ijerph, 2022, doi:10.3390/ijerph191912744_

Round 1

Reviewer 1 Report

The authors searched for articles in MEDLINE and Cochrane Reviews that appeared between 2020 and 2022. This is somewhat of a limited search. Did the authors invoke a snowballing approach in which they pursued other articles that appear in the references of the articles from the MEDLINE and Cochrane Reviews search?

Section 3.5 ends with the statement “Third bullet.” It is not clear what this is supposed to represent.

In Table 3, the authors indicate that the Souza et al study has an observational design. The authors should be more specific as to whether this is a cross-sectional, cohort, or a case-control study design.

The authors searched for articles in MEDLINE and Cochrane Reviews that appeared between 2020 and 2022. This is somewhat of a limited search. Did the authors invoke a snowballing approach in which they pursued other articles that appear in the references of the articles from the MEDLINE and Cochrane Reviews search?

Section 3.5 ends with the statement “Third bullet.” It is not clear what this is supposed to represent.

In Table 3, the authors indicate that the Souza et al study has an observational design. The authors should be more specific as to whether this is a cross-sectional, cohort, or a case-control study design.

Author Response

Thank you very much for Your helpful review. We addressed the comments and made corrections:

1) „The authors searched for articles in MEDLINE and Cochrane Reviews that appeared between 2020 and 2022. This is somewhat of a limited search. Did the authors invoke a snowballing approach in which they pursued other articles that appear in the references of the articles from the MEDLINE and Cochrane Reviews search?”

COM: As a part of the extended research, the references of Lages et al. were scanned and through snowballing effect no more studies, that meet our inclusion criteria, could be identified. 

2) „Section 3.5 ends with the statement “Third bullet.” It is not clear what this is supposed to represent.”

COM: autocorrect error

3) „In Table 3, the authors indicate that the Souza et al study has an observational design. The authors should be more specific as to whether this is a cross-sectional, cohort, or a case-control study design.”

COM:Yes, it's classified as cross-sectional design

Reviewer 2 Report

Wdowiak-Szymanik et al conducted a literature review on the crossing of Covid19 and oral health. The introduction, although short, provides sufficient information. It states that the aim is to show how the pandemic and lockdowns influenced adults' oral hygiene.  The following pages are describing how the authors narrowed down their search for papers on the subject, with too much detail on the types of papers, and the filters, the reasons for exclusions, etc.  In my opinion, this is too much information that perhaps can be in supplementary pages.  The real review, the meat of the information, really is in the current pages 8 and 9 of 13, which is only a page and a half, too short.  I encourage the authors to read the 11 papers the included in their review and try to write themes of those findings, with some graphs that brings all that data together. In addition, the authors should also explore potential new directions of research in a few areas within those themes. It sounds like the authors were trying to write a primary literature paper using publications, but they state that this is a review. 

Author Response

Thank you very much for Your helpful review. We addressed the comments and made corrections

  1. The following pages are describing how the authors narrowed down their search for papers on the subject, with too much detail on the types of papers, and the filters, the reasons for exclusions, etc.  In my opinion, this is too much information that perhaps can be in supplementary pages. 

We reduced the amount of detail and focused on the most important inclusion criteria.

2) The real review, the meat of the information, really is in the current pages 8 and 9 of 13, which is only a page and a half, too short.  I encourage the authors to read the 11 papers the included in their review and try to write themes of those findings, with some graphs that brings all that data together. In addition, the authors should also explore potential new directions of research in a few areas within those themes.

In our opinion, we have included all important informations from the selected 11 articles and listed them in Table 4.

Reviewer 3 Report

Thank you for this interesting  paper to review. I would like to comment it as a well-prepared paper, the prisma flow diagram is also very detailed.

I would only add some aspects to the discussion:

1. The presece of oral manifestations of the disease, including those after the vactination: 

Paradowska-Stolarz AM. Oral manifestations of COVID-19: Brief review. Dent Med Probl. 2021 Jan-Mar;58(1):123-126. doi: 10.17219/dmp/131989. PMID: 33590976.

Mazur M, Duś-Ilnicka I, Jedliński M, Ndokaj A, Janiszewska-Olszowska J, Ardan R, Radwan-Oczko M, Guerra F, Luzzi V, Vozza I, Marasca R, Ottolenghi L, Polimeni A. Facial and Oral Manifestations Following COVID-19 Vaccination: A Survey-Based Study and a First Perspective. Int J Environ Res Public Health. 2021 May 7;18(9):4965. doi: 10.3390/ijerph18094965. PMID: 34066995; PMCID: PMC8125066.

2. The potential use of saliva in the diagnostics

Duś-Ilnicka I, Krala E, Cholewińska P, Radwan-Oczko M. The Use of Saliva as a Biosample in the Light of COVID-19. Diagnostics (Basel). 2021 Sep 26;11(10):1769. doi: 10.3390/diagnostics11101769. PMID: 34679467; PMCID: PMC8534561.

3. The reasons for dental checkups during the pandemic:

Lewandowska M, Partyka M, Romanowska P, Saczuk K, Lukomska-Szymanska MM. Impact of the COVID-19 pandemic on the dental service: A narrative review. Dent Med Probl. 2021;58(4):539–544. doi:10.17219/dmp/137758

Al-Bayaty FH, Baharudin N, Abu Hassan MI. Impact of dental plaque control on the survival of ventilated patients severely affected by COVID-19 infection: An overview. Dent Med Probl. 2021;58(3):385–395. doi:10.17219/dmp/132979 

4. Dental anxiety of the patients during COVID-19:

Berberoğlu B, Koç N, Boyacioglu H, et al. Assessment of dental anxiety levels among dental emergency patients during the COVID-19 pandemic through the Modified Dental Anxiety Scale. Dent Med Probl. 2021;58(4):425–432. doi:10.17219/dmp/139042

Las but not least, Please, add limitations of the study as a separate section.

Thank you in advance.

Author Response

Thank You very much for helpful and very interesting comments with literature recommendations.

1) I would only add some aspects to the discussion:

We used Your literature recommendations in our discussion. We also believe that they enrich the discussion with interesting information and topics.

2) Las but not least, Please, add limitations of the study as a separate section.

We made a separate section.

Round 2

Reviewer 2 Report

Improved from previous draft.